# Biogeochemical Activity of Methane-Related Microbial Communities in Bottom Sediments of Cold Seeps of the Laptev Sea

**DOI:** 10.3390/microorganisms11020250

**Published:** 2023-01-19

**Authors:** Alexander S. Savvichev, Igor I. Rusanov, Vitaly V. Kadnikov, Alexey V. Beletsky, Elena E. Zakcharova, Olga S. Samylina, Pavel A. Sigalevich, Igor P. Semiletov, Nikolai V. Ravin, Nikolay V. Pimenov

**Affiliations:** 1Winogradsky Institute of Microbiology, Research Center of Biotechnology, Russian Academy of Sciences, 119071 Moscow, Russia; 2Institute of Bioengineering, Research Center of Biotechnology, Russian Academy of Sciences, 119071 Moscow, Russia; 3Pacific Oceanological Institute, Russian Academy of Science, 41 Baltiiskaya Street, 690022 Vladivostok, Russia

**Keywords:** Laptev Sea, marine sediments, cold methane seep, microbial community, rates of microbial processes, carbon stable isotope, anaerobic oxidation of methane, pyrosequencing

## Abstract

Bottom sediments at methane discharge sites of the Laptev Sea shelf were investigated. The rates of microbial methanogenesis and methane oxidation were measured, and the communities responsible for these processes were analyzed. Methane content in the sediments varied from 0.9 to 37 µmol CH_4_ dm^−3^. Methane carbon isotopic composition (δ^13^C-CH_4_) varied from −98.9 to −77.6‰, indicating its biogenic origin. The rates of hydrogenotrophic methanogenesis were low (0.4–5.0 nmol dm^−3^ day^−1^). Methane oxidation rates varied from 0.4 to 1.2 µmol dm^−3^ day^−1^ at the seep stations. Four lineages of anaerobic methanotrophic archaea (ANME) (1, 2a–2b, 2c, and 3) were found in the deeper sediments at the seep stations along with sulfate-reducing *Desulfobacteriota*. The ANME-2a-2b clade was predominant among ANME. Aerobic ammonium-oxidizing *Crenarchaeota* (family *Nitrosopumilaceae*) predominated in the upper sediments along with heterotrophic *Actinobacteriota* and *Bacteroidota*, and mehtanotrophs of the classes *Alphaproteobacteria* (*Methyloceanibacter*) and *Gammaproteobacteria* (families *Methylophilaceae* and *Methylomonadaceae*). Members of the genera *Sulfurovum* and *Sulfurimonas* occurred in the sediments of the seep stations. Mehtanotrophs of the classes *Alphaproteobacteria* (*Methyloceanibacter*) and *Gammaproteobacteria* (families *Methylophilaceae* and *Methylomonadaceae*) occurred in the sediments of all stations. The microbial community composition was similar to that of methane seep sediments from geographically remote areas of the global ocean.

## 1. Introduction

Emission of methane from the sediments of the Arctic seas [1], its sources, and the effect of elevated temperature on methane amounts released from marine sediments [2] have been widely discussed. Methane release from the sediments is known to result in the formation of microbial communities using it as a trophic resource [3,4,5]. Since the scale of methane release from marine sediments may vary by several orders of magnitude, the composition of these microbial communities may also differ [6,7,8].

High rates of anaerobic oxidation of methane (AOM) were revealed in subsurface sediment horizons of most marine methane seeps, while oxidation of reduced sulfur compounds occurred at the surface [9]. AOM is carried out by consortia of anaerobic methanotrophic archaea (ANME) and sulfate-reducing bacteria (SRB). ANME belong to the order “*Ca*. Methanophagales” (ANME-1) and to several *Methanosarcinales* lineages –ANME (1, 2a–2b, 2c, and 3) [7,9]. SRB of these consortia are usually *Deltaproteobacteria* closely related to *Desulfosarcina* and *Desulfococcus*, and *Desulfobulbus* [10]. The role of classical aerobic methanotrophs (*Gammaproteobacteria*) is usually considered insignificant due to low availability of oxygen in marine sediments associated with methane seeps [9]. The sites of active mud volcanoes, where rapidly growing aerobic methanotrophs predominate in the oxic upper sediment layers, are exceptional in this respect [11]. Gammaproteobacteria of the order *Methylococcales* are the typical aerobic methanotrophs of such sediments [12,13,14]. Active emission of methane bubbles from marine sediments was observed at the Laptev Sea shelf. The first mapping of the Laptev Sea methane seeps was carried out during the 2008 International Expedition to Siberian Shelf, on board RV *Yakov Smirnitskii* [15]. Subsequent studies revealed over 100 local fields of gas seepage at the depths of 50–90 m [16].

Apart from detection of methane sources, geophysical studies of methane emission from the Laptev Sea sediments are aimed to determine the relations between the scale of methane emission and the degradation of deep gas hydrates [2]. Investigation of the benthic communities with trophic dependence upon the activity of methanotrophic and chemotrophic bacteria is another area of research [17].

The first data on microbial community composition and the rates of the methane cycle microbial processes in the Laptev Sea sediments and near-bottom water were obtained were obtained for a local 50 × 50 m polygon at the depth of 71 m [18]. The high methane oxidation rate in the sediments was shown to result from activity of a consortium of ANME-2 a/b archaea and sulfate-reducing bacteria. Bacterial mats visible on the sediment surface within the gas seepage field were found to be predominated by bacteria of the genera *Sulfurovum* and *Arcobacter*.

The goal of the present work was to obtain information on the rates of microbial processes of the carbon (methane) and sulfur cycles in the upper sediments at methane discharge sites of the Laptev Sea shelf zone and on the composition of microbial communities involved in the geochemically significant processes. This goal was achieved by integration of the biogeochemical methods using radiolabeled compounds and molecular genetic techniques to reveal the taxonomic composition of microbial compounds.

## 2. Materials and Methods

### 2.1. Field Study and Sampling

Samples of the bottom sediments were collected in October 2018 in the Laptev Sea during the 73rd cruise of RV *Akademik Mstislav Keldysh*. Upper sediments were collected at three most representative stations at the shelf edge (Figure 1). Two of them (AMK73-6027 and AMK73-6045) were located at the sites of methane seeps, while station AMK73-6053, not affected directly by the river flow or methane seeps, was used as the reference one.

At each station, 6–7 sediment horizons (from the surface to 20–30 cm deep) were investigated. The samples were obtained with a plastic tube (15 cm in diameter) inserted into the sample collected with a box corer grab immediately after its retrieval on board the ship. Samples of the sediment layers were prepared in the onboard laboratory for radiotracer, chromatographic, biogeochemical, microbiological, and molecular genetic research. All experiments with the sediments were carried out within several hours after sampling at the temperatures close to in situ values. The samples for DNA isolation were collected from the same horizons, frozen, and stored at –20 °C.

To determine methane content, the head-space method of sampling was used. Methane concentration was measured by the phase-equilibrium degassing method on a Kristall-2000-M gas chromatograph (Russia) equipped with a flame ionization detector. The measurement error did not exceed ±5%.

Pore water was obtained from the sediments by centrifugation at 5000× *g* on a TsUM-1 centrifuge (Russia). The alkaline reserve was determined with the relevant reagent kit (Merck, Germany). The concentrations of sulfate and chloride ions in pore water were determined on a Staier ion chromatograph (Russia).

The rates of microbial processes: dark CO_2_ assimilation (DCA), hydrogenotrophic methanogenesis (MG), methane oxidation (MO), and sulfate reduction (SR) was determined by radiotracer analysis with ^14^C- and ^35^S-labeled substrates. For this purpose, bottom sediment samples (2.5 cm^3^) were collected with a cut-off syringe with a rubber plunger and sealed with a gas-tight butyl rubber stopper. The labeled substrate (0.2 mL) was injected with a tuberculin syringe by piercing the stopper at the center and distributing the substrate uniformly along the syringe length. Methane oxidation rate was determined with ^14^C-labeled methane dissolved in sterile distilled water (1 μCi per sample). Sulfate reduction rate was determined with ^35^S-labeled sulfate (2.5 μCi per sample). The rates of methanogenesis and microbial CO_2_ assimilation were determined using ^14^C-labeled bicarbonate (4.0 μCi per sample). The samples were incubated for 24 h at the temperature close to in situ values and then fixed with 1.0 mL of 1 M KOH. Sediment samples fixed with KOH prior to addition of the labeled substrates were used as the controls. Subsequent sample processing and calculation of the rates of microbial processes were carried out as described previously [19]. Radioactivity (^14^C and ^35^S) of the products of microbial processes was measured on a PackardTRI-CarbTR 2400 liquid scintillation counter (USA). Numerical values of the rates of microbial processes (DCA, MG, MO, and SR) were calculated using the averages for two replicate measurements per each sample.

C_org_ content in the sediments was determined after removal of carbonates on an AN-7560 express analyzer (Russia) by registering the amount of CO_2_ released after incineration of the sample at ~900 °C in the flow of CO_2_-free air.

### 2.2. Determination of the Composition of Microbial Communities by High-Throughput Sequencing of the 16S rRNA Genes

The total DNA was extracted from sediment samples using Power Soil DNA isolation kit (MO BIO Laboratories, Inc., Carlsbad, CA, USA). PCR amplification of 16S rRNA gene fragments comprising the V3–V6 variable regions was carried out using the universal prokaryotic primers PRK 341F (50-CCTAYGGGDBGCWSCAG) and PRK 806R (50-GGACTACNVGGGTHTCTAAT) [20]. The PCR fragments were bar-coded using the Nextera XT Index Kit v.2 (Illumina, San Diego, CA, USA) and purified using Agencourt AMPure beads (Beckman Coulter, Brea, CA, USA). The concentrations of PCR products were calculated using the Qubit dsDNA HS Assay Kit (Invitrogen, Carlsbad, CA, USA). All PCR fragments were then mixed and sequenced on Illumina MiSeq (2 × 300 nt from both ends). Pairwise overlapping reads were merged using FLASH v.1.2.11 [21]. The final dataset consisted of 2,198,811 16S rRNA gene reads (Table 1).

All sequences were clustered into operational taxonomic units (OTUs) at 97% identity using the USEARCH v. 11 program [22]. Low-quality reads and chimeric sequences were removed by the USEARCH algorithms. To calculate OTU abundances, all reads obtained for a given sample (including singleton and low-quality reads) were mapped to OTU sequences at a 97% global identity threshold by Usearch. The taxonomic assignment of OTUs was performed by searching against the SILVA v.138 rRNA sequence database using the VSEARCH v. 2.14.1 algorithm [23]. OTUs assigned to chloroplasts, mitochondria, and eukaryotes, as well as OTUs containing only one read in the entire dataset and likely resulting from sequencing errors, were excluded from the analysis.

The Chao1 and Shannon diversity indices at a 97% OTU cut-off level were calculated using Usearch v.11 [22]. To avoid sequencing depth bias, the numbers of reads generated for each sample were randomly sub-sampled to the size of the smallest set (reads from 6840 sample) using the “otutab_rare” command of Usearch.

Illumina technology (HiSeq2500) was used for metagenome sequencing. TruSeq DNA library was prepared using the Nextera DNA Library preparation kit; its sequencing on the Illumina HiSeq2500 was performed using the HiSeq Rapid Run v2 sequencing reagents kit in the format of paired reads (2 × 150 nt). A total of 31,740,971 read pairs (~9.5 Gb) and 70,291,374 read pairs (~21 Gb) were generated for the samples st6027 and st6045, respectively. The adapter sequences were removed using Cutadapt v. 1.14, and low quality read ends (Q < 30) were trimmed with Sickle v. 1.33. Reads were de novo assembled int us-ing MEGAHIT v.1.2.9. Contigs were binned into clusters representing metagenome-assembled genome (MAGs) using MetaBAT v.2.15. The completeness of the MAGs and their possible contamination (redundancy) were estimated using CheckM v.1.1.3. The taxonomic classifications were assigned to the MAGs using the Genome Taxonomy Database Toolkit (GTDB-Tk) v.1.3.0. Functional and metabolic pathway annotations were done with METABOLIC v.4.0 software and additional Diamond v.2.0 homology search against the Uniref database.

The raw data generated from the 16S rRNA gene and metagenomes sequencing were deposited in the NCBI Sequence Read Archive (SRA) and are available via the BioProject PRJNA679168.

### 2.3. Stable Isotope Analyses

The δ^13^C-C_org_ isotopic composition of the bottom sediments was determined in FRC Biotechnology. The preliminary treatment of samples was carried out in four-fold-diluted concentrated HCl (1/4) with heating to remove the carbonates. Then, the samples were calcined in a vacuum circulation device installed in quartz tube with CuO at 900 °C. The released CO_2_ was sealed in ampules and used for the isotope analysis. The δ^13^C-C_org_ values were determined using a Delta Plus mass spectrometer (Thermo Electron Corporation, Germany). The results of isotope analysis, δ^13^C, are reported in relation in conventional delta (δ) units relative to the international standard Vienna Pee Dee Belemnite (VPDB). Thus the results of δ^13^C isotope analysis obtained using mass spectrometers in various laboratories have an excellent reproducibility. The carbon isotope composition of methane, δ^13^C-CH_4_, was measured using a TRACE GC gas chromatograph (Thermo Fisher Scientific, Germany) connected to a Delta Plus mass spectrometer in FRC Biotechnology. The measurement error of δ^13^C did not exceed ± 0.1‰.

## 3. Results

### 3.1. Geochemical Characterization of the Upper Sediments (0–27 cm)

At the zone of methane seeps (st. 6027 и 6045), the upper sediments were represented by black or dark brown, usually fine aleuric (less often aleuric), slightly oxidized, or oxidized deposits (Eh from +120 to −60 mV) (Table 1; Figure 2).

**Table 1 microorganisms-11-00250-t001:** General properties of the sediments and number of read sequences of 16S rRNA at three Laptev Sea sites.

Station	Horizon	Description	Eh, mV	C_org_, %	CH_4,_ µM L^−1^	Alk, µM L^−1^	Reads Initial
6027 (seep) 76.89 N 127.80 E	0–1 cm	Oxidized, strongly hydrated, brown	+200	0.49	1.04	3.0	140504
	1–3 cm	Oxidized, sandy-aleuric, gray	+80	0.62	1.31	2.8	110552
	3–5 cm	Transitional, gray, and black, of average density	+20	0.52	1.34	3.2	118701
	5–8 cm	Reduced, aleurite, black, dense	−60	0.41	1.59	3.8	111650
	8–14 cm	Reduced, black, more dense	−110	0.43	1.05	5.0	130303
	14–18 cm	Reduced, black, soft	−100	0.51	1.82	7.7	110552
6045 (seep) 76.77 N 125.76 E	Near-bottom water		+220		0.17	2.1	123661
	Warp	Light brown	+160		0.35	2.3	120119
	0–1 cm	Reduced, brown, very liquid	+140	0.98	2.58	3.0	111475
	1–3 cm	Slightly oxidized, gray, liquid	+60	1.05	6.27	3.0	106337
	3–7 cm	Transitional, gray, and black, of average density	−10	0.88	13.17	4.0	131133
	7–12 cm	Reduced, aleurite, black, dense	−120	0.97	20.32	5.4	132668
	12–18 cm	Reduced, black, more dense	−160	0.90	36.73	7.0	108153
	18–23 cm	Reduced, aleurite, black, dense	−160	0.53	16.92	7.5	123661
	23–27 cm	Reduced, aleuric sand, black	−110	0.39	15.88	8.5	120119
6053 (reference) 76.74 N 128.45 E	0–3 cm	Reduced, brown	+180	1.47	0.93	2.5	121114
	3–6 cm	Reduced, light brown, or gray	+120	1.26	1.10	2.6	124989
	6–10 cm	Transitional, gray, more dense	+30	1.21	1.38	3.2	126268
	10–16 cm	Gray, with hydrotroilite inclusions	−10	1.22	1.97	3.4	120495
	16–23 cm	Dark gray, hydrotroilite	−40	1.08	1.63	3.2	124246
	23–27 cm	Dark gray, hydrotroilite	−40	1.23	1.60	3.4	136443

Immediately below the thin surface layer, the sediments of the seep areas were strongly reduced (Eh from −50 to −160 mV). The sediment of the reference st. 6053 was oxidized at the surface and slightly reduced in the lower horizons (Eh from +100 to −40 mV). In all cases, the temperature of the sediments varied from −1.2 to −0.5 °C. The alkaline reserve of pore water (Alk) at stations 6027 and 6045 was up to 7.7–8.5 mg eq L^−1^ in the 14–18 and 23–27 cm layers, respectively. At the reference station, Alk of the pore water did not exceed 3.4 mg eq L^−1^.

Methane content in the upper horizons of the studied sediments varied from 0.9 to 2.6 µmol CH_4_ dm^−3^. Methane content increased with depth to 1.8 µmol CH_4_ dm^−3^ (18 cm) in the core of the st. 6027 sediments, and to 36.7 µmol CH_4_ dm^−3^ at st. 6045 (18 cm). Methane content at st. 6053 (18 cm) was 1.6 µmol CH_4_ dm^−3^ (Figure 3).

Together with other physicochemical parameters, the data on methane concentrations indicated that the sediment of st. 6045 came from the seep zone, st. 6027 was at the periphery of the seep zone, and at st. 6053, the sediment was collected outside the methane discharge zone.

The ratio between the concentrations of sulfate and chloride ions (SO_4_^2−^/Cl^−^ × 1000) in pore water of marine sediments is a reliable indicator of the geochemical consequences of microbial sulfate reduction. This coefficient is stable in seawater (139.6) [24]. The relative abundance of sulfate ion in pore water of stations 6027 and 6045 decreased gradually with depth, indicating partial consumption of this ion in the course of sulfate reduction (Figure 4).

Pore water of st. 6053 (the reference station) showed a different tendency. The relative abundance of sulfate ion increased slightly with depth, indicating the absence of the effect of sulfate reduction. Elevated concentrations of sulfate ion were observed at st. 6027 in the 2–4 cm horizon, which was probably due to oxidation of reduced sulfur compounds.

### 3.2. Composition of the Microbial Community of the Sediments

A total of 2,198,811 16S rRNA gene fragments were identified in order to characterize the composition of microbial communities. Clusterization of these sequences resulted in the identification of 14,261 bacterial and 4563 archaeal OTU at 97% identity. The alpha diversity indices showed high bacterial and lower archaeal diversity in all sediment samples (Table 2).

Archaea constituted 27.3% to 71.8% of all 16S rRNA gene sequences (Figure 5). They were represented by 11 phyla, as determined using SILVA according to the GTDB database: *Aenigmarchaeota*, *Altiarchaeota*, *Asgardarchaeota*, *Crenarchaeota*, *Euryarchaeota*, *Halobacterota*, *Hydrothermarchaeota*, *Iainarchaeota*, *Micrarchaeota*, *Nanoarchaeota*, and *Thermoplasmatota*.

Most archaea revealed at the seep stations belonged to the phylum *Halobacterota*, order *Methanosarciniales* of four ANME methanotrophic lineages (1, 2a–2b, 2c, and 3). ANME of groups 2a–2b were predominant, at up to 46.3% of the whole microbial community.

Aerobic ammonium-oxidizing *Crenarchaeota* of the family *Nitrosopumilaceae* were predominant in the upper sediment layers at all stations. The class *Bathyarchaeia* predominated in anoxic sediment layers of the reference station, while their relative abundance at the seep stations was up to 3%. The order *Woesearchaeales* (phylum *Nanoarchaeota*) was predominant at all stations in the 3 – 16 cm layer. The phylum *Asgardarchaeota* was represented by three classes, *Heimdallarchaeia*, *Lokiarchaeia*, and *Odinarchaeia,* and was detected only in oxic layers. Among them, the most abundant ones belonged to the classes *Lokiarchaeia* (up to 18.55%) and *Heimdallarchaeia* (up to 6.78%). The closest relatives of these organisms have been revealed in marine sediments of the Bay of Aarhus (Denmark), Skan Bay (Alaska), the Gulf of Mexico, and the Sea of Japan. Methanogenesis is usually limited in marine sediments where sulfate is present since sulfate-reducing bacteria have a higher affinity to hydrogen and acetate [25]. Such C1 compounds as methanol and methylamines (methylamine, dimethylamine, and trimethylamine) or sulfur compounds (methanethiol and dimethyl sulfide) may be used by some methanogens as the substrates for growth and methanogenesis. Almost no such archaea were detected at the seep stations, while *Methanococcoides* (11–6.8% in deep layers) and *Methanomassiliicoccales* (1.8–3.2%, also in deep layers), potentially capable of using these substrates for methanogenesis, were revealed at the reference station. The share of MBGD archaea at the seep stations increased with depth, while at reference station 6053, they predominated only at depths from 3 to 10 cm. A significant part of archaea (up to 8%) was represented by new lineages, for which the metabolic pathways are as yet unknown.

The phyla *Actinobacteriota* and *Bacteroidota*, comprising heterotrophic bacteria capable of degrading various organic compounds, were predominant in the surface horizons of all stations. Members of the genera *Sulfurovum* and *Sulfurimonas*, belonging to the phylum *Campylobacterota* (previously assigned to the class *Epsilonproteobacteria*), were also revealed in the upper sediments of the seep stations. These are chemolithoautotrophic bacteria able to oxidize sulfur-containing compounds. At all stations, the share of the phylum *Chloroflexi* increased with depth; it was represented by uncultured groups, which have been detected in various marine seeps. Small numbers of methanotrophic bacteria were found at all stations; they belonged to the classes *Alphaproteobacteria* (*Methyloceanibacter*) and Gammaproteobacteria (families *Methylophilaceae* and *Methylomonadaceae*).

The phylum *Desulfobacterota* was detected at all stations; it predominated in the medium sediment layers. Most of these organisms were represented by uncultured members of the families *Desulfosarcinaceae* (genera SEEP-SRB1 and LCP-80) and *Desulfobulbaceae*, which have been revealed in various bottom sediments.

### 3.3. Analysis of Microbial Genomes Assembled from Metagenomes

Sequencing of metagenomes of the microbial communities of the seep stations 6027 (14–18 cm) and 6045 (18–23 cm) made it possible to assemble 19 and 28 MAGs, respectively. The taxonomic position of these MAGs was determined by phylogenetic analysis using concatenated sequences of conservative marker genes in Genome Taxonomy Database [26]. Taxonomic classification revealed the same major bacterial lineages as those revealed by the 16S rRNA gene profiling.

Four genomes of the phylum *Desulfobacterota* and four genomes of the phylum *Chloroflexota* were assembled for st. 6027. Among *Desulfobacterota*, two MAGs (bin40 and bin18) were assigned to the order *Desulfobacterales*, while the other two (bin2 and bin25) belonged to the uncultured order C00003060, corresponding to the SEEP-SRB1c lineage (Figure 6a). The closest relatives of the order C00003060 were found only in hydrocarbon-rich marine deposits, such as Hydrate Ridge cold seeps of the Pacific coast [27]. Uncultured *Zixibacteria* were represented in the metagenome by two MAGs. Archaea were represented by bin4 and were assigned to the ANME-2c group of AOM (anaerobic oxidizers of methane).

A total of 21 MAGs assembled from st. 6045 metagenome were assigned to Bacteria, including the phyla *Desulfobacterota* (3 MAG), *Chloroflexota* (5 MAGs), and *Zixibacteria* (2 MAG). Four archaeal genomes represented ANME lineages, ANME-2a-2b, ANME-2c, and ANME-1, with bin 53 (ANME-2a-2b) comprising 3.05% of the whole metagenome. Bin 53 was closely related (average amino acid sequence identity (AAI) of 95.79%) to archaeon HR1 ASM292619v1 found in a bottom methane seep in the Pacific Ocean and assigned to the ANME-2 group. Genome analysis revealed all the key genes of reverse hydrogenotrophic methanogenesis in three MAGs, st. 6027_bin4, st6045_bin50, and st6045_bin53. The genes *mcr*, *mtr*, *mtd*, *ftr*, and *fmd* were not found in st6045_bin61, while *mer*, *mtd*, and *mch* did not occur in st6045_bin31. Probably, the absence of these genes is due to the incompleteness of the assembled MAGs. Other archaeal MAGs from st. 6045 represented the phylum *Asgardarchaeota* (3 genomes) (Figure 6b).

### 3.4. Genomes of Sulfate-Reducing Bacteria

Among sulfate-reducing bacteria, members of three phyla, *Nitrospirota* (order *Thermodesulfovibrionales*), *Desulfobacterota* (orders C00003060, *Desulfobacterales*, and *Desulfobulbales*), and *Zixibacteria*, were revealed in the metagenomes. All of them contained complete sets of genes responsible for sulfate reduction. Five MAGs of sulfate-reducing bacteria were assembled in the metagenomes from St. 6027: 3 MAGs belonged to *Desulfobacterota*, and 2 MAGs to *Zixibacteria*. Three MAGs comprising main genes of dissimilatory sulfate reduction pathway were assembled from the metagenome of st. 6045: two MAGs belonging to *Desulfobacterota* and one to *Zixibacteria*.

### 3.5. Zixibacteria, a Possible Partner of ANME Archaea in Methane Seep Microbial Communities

The phylum *Zixibacteria* was first described as the RBG-1 lineage (Rifle Back Ground organism (RBG)) in the subsurface aquifer sediments [28]. Sequencing of two methane seep metagenomes resulted in assembly of 4 genomes of *Zixibacteria* belonging to the class MSB-5A5. One of the MAGs, Bin19, was assembled into 171 contigs with the total length of 4,504,648 bp. According to the CheckM estimates, this MAG is 100% complete with 3.3% possible contamination. Analysis of the Bin19 genome revealed the 5S-23S-tRNA(Ala)-tRNA(Ile)-16S rRNA operon and 41 tRNA genes. A GenBank search revealed that the closest relatives of Bin19 were found in sediments and hydrothermal seeps, but the 16S rRNA gene sequence identity was only 91%. While the genome annotation led to the prediction of 4033 potential protein-coding genes, the functions of only 1344 of these could be hy-pothesized by comparison with the NCBI databases.

Search for the phylogenetically related microorganisms based on genome similarity revealed that the closest of Bin19 is *Zixibacteria* sp. AS27yjCOA_31 (GCA_012797915), with 60.84% AAI. It was detected in the metagenome of an anaerobic digester fermenting organic waste [29], which exhibited 60.84% identity of amino acid sequences (AAI). According to the criteria proposed for the determination of the phylogenetic position of uncultured microorganisms [30], Bin19 and candidate division *Zixibacteria* sp. AS27yjCOA_31 belonged to different species of the same genus.

To analyze *Zixibacteria* phylogeny, a phylogenetic tree was constructed based on concatenated sequences of 43 conservative marker genes comprising Bin19, Bin47, Bin10, Bin45, and other available *Zixibacteria*. Our results (Figure 7) confirmed that candidate division *Zixibacteria* sp. AS27yjCOA_31 was the closest relative of Bin19.

Analysis of the Bin19 genome revealed the genes encoding all enzymes of glycolysis, the nonoxidative branch of the pentose phosphate pathway, and the TCA cycle. Unlike the previously described RBG-1 genome [28], it contained a complete set of genes encoding the flagellum. No pathways for nitrate and sulfate reduction were found in the RBG-1 genome. Bin19 bacterium was predicted to be capable of sulfate reduction, and all the relevant genes were found in the genome. The genomes of RBG-1 and Bin19 code also the complete oxidative phosphorylation pathway, which may enable aerobic respiration of these organisms. The presence of numerous protease genes indicated the ability of Bin19 to hydrolyze various proteins.

Thus, *Zixibacteria* is metabolically versatile, which may explain the significance of this group in marine deposits, where they are often detected. Their ability to reduce sulfate may be coupled to methane oxidation by ANME archaea.

### 3.6. Rates of Microbial Processes

Dark CO_2_ assimilation (DCA) provides for the most complete characterization of the rate of heterotrophic and chemotrophic processes in the sediments.

In all studied sediments, DCA reached its maximum in the surface layers and decreased with depth (Figure 8). The absolute DCA value at the reference station was approximately half of that for the methane seep stations. The share of carbon in microbial biomass was 18–35% in the surface horizons (68% in the warp of st. 6045) and only 2–8% in deep horizons. Such distribution of assimilated carbon indicated a stable decrease in the efficiency of cell metabolism and in the cell division rate.

The rates of hydrogenotrophic methanogenesis (MGh) were rather low in all studied sediments (Figure 9).

The absolute values of MGh rates in the studied sediments were low (0.4–5.0 nmol dm^−3^ day^−1^) and similar for all stations (the methane seep zone, its periphery, and the reference zone outside the seep). The 6–10 cm layer of st. 6053, with the rate of 12.0 nmol dm^−3^ day^−1^, was somewhat exceptional in this respect. Thus, MGh occurring in the studied sediment layers contributed insignificantly to the total methane content (Figure 3). In the sediments of st. 6027, the rate of methane generation regularly increased with depth, indicating better conditions for the activity of hydrogenotrophic archaea in deeper horizons.

The absolute values of MO rates in the sediments of all three stations were significantly higher than the rates of methanogenesis (Figure 10). This indicated that most methane was delivered to the upper sediment horizons from lower layers. Active MO was observed for the sediments of st. 6045 (from 400 to 1200 nmol dm^−3^ day^−1^), where the layers with high and low MO activity were located without visible regularity. MO rates in the sediments of the seep periphery (st. 6027) were lower (from 55 to 350 nmol dm^−3^ day^−1^). While MO rates in the sediments of the reference st. 6053 were even lower (from 30 to 130 nmol dm^−3^ day^−1^), they were still higher than the rates of archaeal methanogenesis (Figure 9). Microbial methane oxidation results in the oxidation of methane carbon to CO_2_, as well as its incorporation into microbial biomass and dissolved organic matter. It should b noted that in the sediments of the active methane seep, the share of methane carbon in dissolved organic matter regularly increased with depth (from 21% to 71%). The share of methane carbon in the cell biomass was low (0.5% to 2.5%) and always decreased with depth.

The rates of sulfate reduction in the sediments of two seep stations, st. 6027 and st. 6045, were similar: from 1.5 to 6.0 µmol dm^−3^ day^−1^ (Figure 11). The SR rate peaked in the 3–7 cm horizon and regularly decreased in deeper layers. The SR rate in the sediments of the reference station was five times lower (from 0.3 to 1.0 µmol dm^−3^ day^−1^). Sulfate reduction resulted in a reduction of sulfate sulfur to sulfide (55 to 80%) and to combined pyrite, elemental, and cellular sulfur. No visible patterns were revealed in the distribution of reduced sulfur species at different stations and in different horizons.

### 3.7. Carbon Isotopic Composition of Organic Matter and Methane

The carbon isotopic composition of organic matter (OM) was measured for the sediments from all three stations. The spread of the values for different horizons varied within 2–3‰ (Table 3). The average values, however, revealed that the lightest OM carbon(δ^13^C_org_ = −29.6‰) was present in the sediments of the active seep (st. 6045). The heaviest OM carbon isotopic composition was found in the sediments of the background station 6053 (δ^13^C_org_ = −26.7‰).

The carbon isotopic composition of methane was determined only for the sediments of st. 6045. The δ^13^C-CH_4_ values varied from −98.9% to −77.6‰, with an average of −83.3‰.

## 4. Discussion

Most works on the investigation of microbial communities of the cold methane seeps concentrate presently on the application of molecular ecological techniques, which provide a detailed description of the taxonomic composition of the communities [31,32,33]. The predominance of such works is, to a significant degree, due to the rapid progress of molecular genetic techniques and the possibility of obtaining data on the presence of various microbial physiological groups. It is, however, evident that the understanding of the role of microbial communities in the biogeochemical processes occurring in cold methane seeps requires information on the rates of microbial processes of the carbon and sulfur cycles. Unfortunately, quantitative studies on the rates of microbial compounds by radiotracer analysis are presently rather few [34,35,36], although the techniques of microbial biogeochemistry based on the application of trace amounts of radiolabeled compounds are well known [37,38,39,40,41]. In the present work, we characterized the microbial communities at the sites of two cold methane seeps and one remote site (background) in the Laptev Sea and determined the rates of the major biogeochemical processes of the carbon and sulfur cycle. The work also included an analysis of two metagenomes from the deep layers of methane seep sediments.

### 4.1. Rates of Microbial Methane Oxidation in the Sediments of the Global Ocean

Methane content in the regular sediments of the Arctic seas (with low OM levels and located outside the zones of hydrocarbon discharge) varies from 0.1 to 10 µmol dm^−3^ (Table 4). Similar or higher methane content was found in the sediments of the Black and South China seas and in regular (outside the methane seep zones) sediments of the Gulf of Mexico. High (10–100 µmol dm^−3^) and very high (up to 3000 µmol dm^−3^) concentrations of dissolved methane were revealed in the sediments of OM-rich shelves, cold methane seeps, pockmarks, mud volcanoes, and geologically active faults. Methane saturating the sediments may be of biogenic (microbial), abiogenic, and mixed origin [42]. Direct measurements in the surface sediment layer in various seas and oceans (Figure 5, Table 4) revealed extremely low (from 5 × 10^−5^ µmol dm^−3^ day^−1^) and low (up to 3.8 µmol dm^−3^ day^−1^) rates of in situ methanogenesis, compared to flooded soils and bogs. High levels of dissolved methane in the surface layers of marine sediments are explained by methane release from lower horizons or from deep reserves. The rates of microbial methane oxidation (both aerobic bacterial and archaeal AOM) in the upper sediments are always higher than the rates of methanogenesis (Table 3). Microbial methane oxidation limits methane release into the water column and acts as a trophic resource for methanotrophic benthic ecosystems. The comparison revealed (Table 2) that methane content in the upper sediments of the seeps investigated in the present work was considerably lower than in the sediments from the zones of active hydrocarbon discharge (Haakon Mosby, Gulf of Mexico, Hikurangi margin cold seeps, etc.). The rates of both methane oxidation and sulfate reduction in the studied Laptev Sea methane-bearing sediments were also noticeably lower than the records for the hydrocarbon discharge zones (Table 3). At the same time, methane oxidation rates in the sediments of the Laptev Sea methane seeps were significantly higher (10 times and more) than in the sediments outside the seep area and in the sediments of most Arctic seas (the Barents, Kara, and Chukchi seas; Table 2). The rate of methane oxidation (MO and AOM) is probably the main factor determining the composition of microbial communities using methane as the major trophic resource. Methane oxidation launches the process of sulfate reduction, and excessive production of sulfide results in the development of visually discernible bacterial mats, where filamentous colorless sulfur bacteria *Beggiatoa* predominate [34,43,44,45]. Predominant bacteria of the visually discernible bacterial mats formed at the surface of the Laptev Sea sediments belong to the genera *Sulfurovum* and *Arcobacter*. Formation of a dense microbial community with the predominance of these bacteria did not require high methane consumption, unlike the *Beggiatoa*-dominated mats.

### 4.2. Carbon Isotopic Composition of Organic Matter in the Sediments

The carbon isotopic composition of methane in the methane seep sediments of st. 6045 had a high abundance of the light isotope ^12^C (δ^13^C-CH_4_ from −98.9‰ to −77.6‰, on average −83.3‰) indicated its modern biogenic (archaeal) origin. Light carbon isotopic composition of OM in the sediments of two methane seeps (average values δ^13^C-C_org_ = −29.6 and −28.2‰) indicated a significant contribution of the methanotrophic microbial community to the formation of ON (Table 3). The difference from the control value (δ^13^C-C_org_ for the sediments of the reference st. 5053) was quite noticeable: δ^13^C-C_org_ = −26.7‰. Since the quality and quantity of both autochthonous and terrigenous OM at all three closely located stations were similar [53], the differences in the carbon isotopic composition of OM in the sediments resulted from different rates of microbial processes of the methane cycle. The studied sediments differed considerably in their OM content (Table 1). In the methane seep sediments, C_org_ values were 0.5% and 0.81%, while the average C_org_ content in the sediments of the reference station was 1.25%. In our opinion, active processes of AOM, as well as sulfate reduction, activate the decomposition of allochthonous and autochthonous OM, which results in its lower content in the sediments.

The carbon isotopic composition of methane in the Laptev Sea methane seep field, Buor-Khaya Bay, and Dmitry Laptev Strait varied within a broad range. The δ^13^C-CH_4_ values varied from −105% to −42.6‰ vs. VPDB [54,55].

### 4.3. Composition of the Microbial Community Responsible for AOM

It was previously established that the anaerobic oxidation of methane by ANME archaea, coupled with sulfate reduction, is the main process occurring in marine cold methane seeps [56,57]. At the seep stations, the relative abundance of ANME archaea increased with depth, unlike the reference station, where they were almost absent. Among the known ANME lineages, the clades ANME-1, ANME-2a-2b, ANME-2c, and ANME-3 were detected. The ANME-2a-2b clade was predominant in the deeper layer, constituting up to 46% of the microbial community according to the 16S rRNA gene profiling data. Apart from ANME archaea, sulfate-reducing deltaproteobacteria (phylum *Desulfobacterota* according to the GTDB taxonomy) were found in all layers of the seep sediments. Among them, the predominant ones belonged to *Desulfobacteraceae*, *Desulfosarcinaceae* (the SEEP-SRB1 group), and *Desulfobulbaceae*. Some of these groups are known as partners of ANME archaea. Thus SEEP-SRB1 revealed in the present work was previously shown to be capable of anaerobic methane oxidation and sulfate reduction in consortia with ANME-2a-2b [10,58].

For a better understanding of microbial processes in methane seeps, two metagenomes from two sediment layers were studied. Although numerous publications deal with metagenomic studies of cold methane seeps [59,60,61,62,63,64,65], a full complex of such studies, including direct radiotracer measurements of the rates of microbial methane oxidation and sulfate reduction, has not been carried out in the Arctic seas. The studied metagenomes contained both ANME archaea and sulfate-reducing *Desulfobacterota*, as well as all the genes responsible for AOM and dissimilatory sulfate reduction. Interestingly, members of the phylum *Zixibacteria* were found in both metagenomes, and their genomes contained all the genes required for sulfate reduction. These microorganisms, which are often found in marine sediments, may probably also interact with ANME archaea, similar to *Desulfobacterota* [28,66,67]. Our data indicate significant similarities between microbial communities of the underwater methane seeps located in geographically remote areas of the Global Ocean. Methanotrophic ANME archaea and sulfate-reducing bacteria form the trophic base of the seep communities; the predominance of specific ANME lineages depends on the physicochemical properties of the sediments and on the scale of gas emission. The physicochemical parameters of the habitats of ANME archaea determining the predominance of specific ANME clades are presently actively discussed in the literature [68,69]. Thus, the ANME-1 group predominates in the mats formed on coral-like structures at the sites of methane seep discharge into the anoxic Black Sea water column [70], in deep layers of Arctic sediments, where methane hydrates appear [71], and under conditions of low methane and sulfate concentrations in the methane-sulfide transition zone [72].

Our results on the occurrence of ANME archaea in subsurface sediment layers at the seep sites in the Laptev and Barents seas indicate the predominance of the ANNE types 2a and 2b. While the 16S rRNA gene sequences of ANME types 1 and 3 were also revealed, their abundance was significantly lower and did not change depending on the sediment horizon.

## Figures and Tables

**Figure 1 microorganisms-11-00250-f001:**
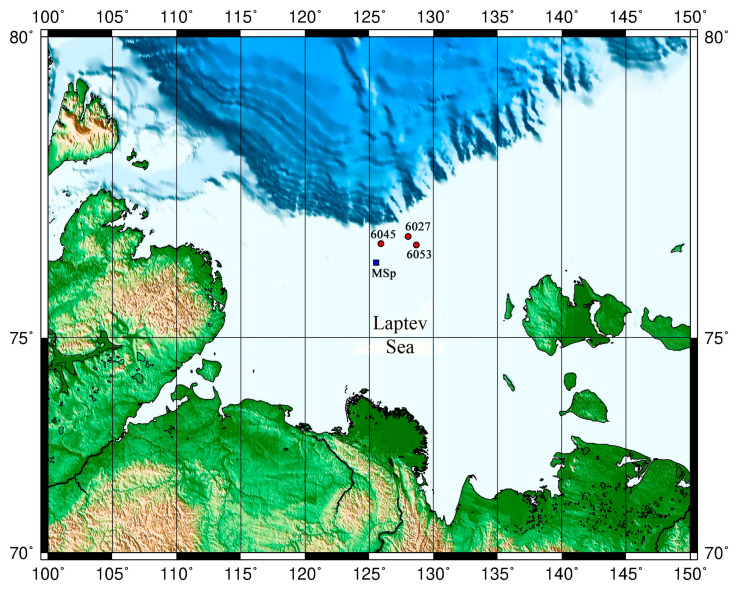
Schematic map of the studied area (Laptev Sea). AMK 6027, 6045, and 6053 are the stations in the shelf edge area. MSp is the methane seep polygon [18].

**Figure 2 microorganisms-11-00250-f002:**
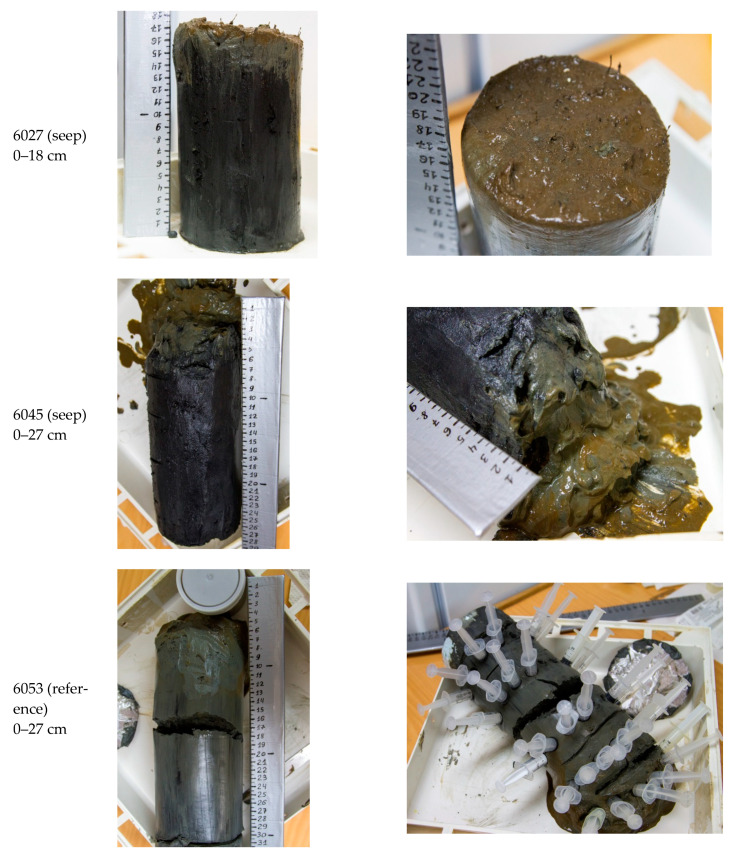
Bottom sediment cores from three stations in the Laptev Sea (73-rd cruise of RV *Akademik Mstislav Keldysh)*.

**Figure 3 microorganisms-11-00250-f003:**
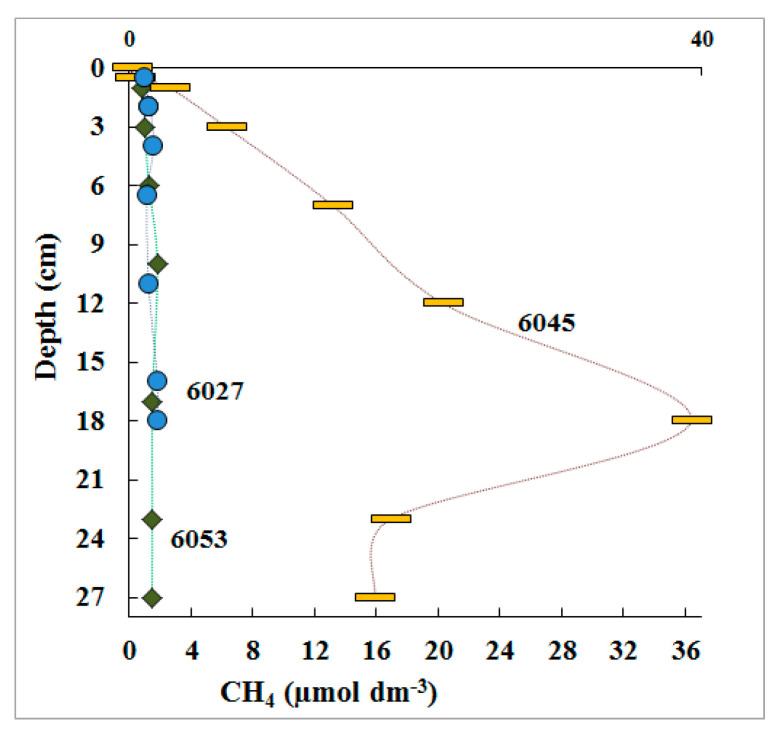
Methane content (CH_4_, µmol dm^−3^) in pore water of three Laptev See shelf sediment samples.

**Figure 4 microorganisms-11-00250-f004:**
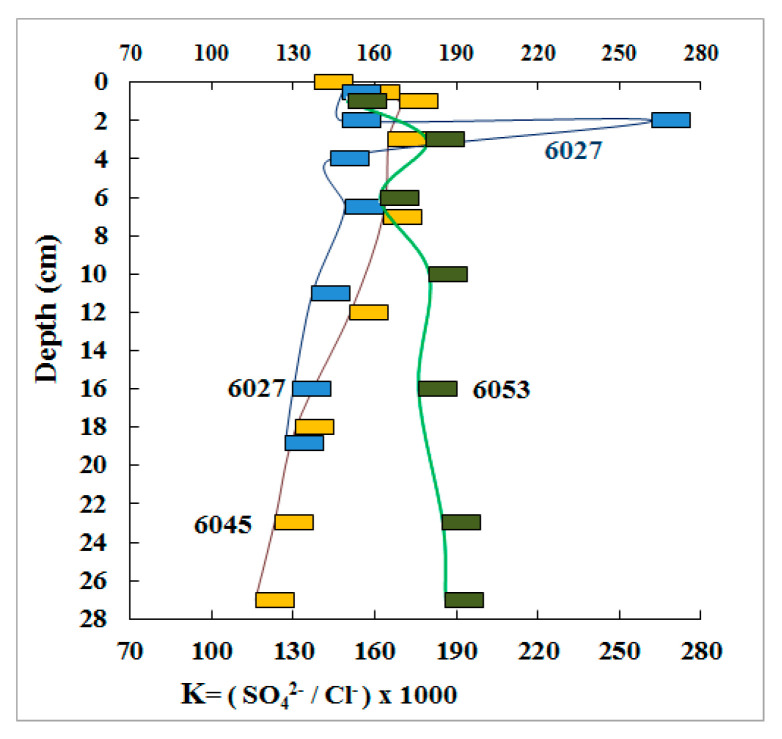
The K coefficient characterizing the ratios between the concentrations of chloride and sulfate ions in pore water of three Laptev Sea shelf sediments.

**Figure 5 microorganisms-11-00250-f005:**
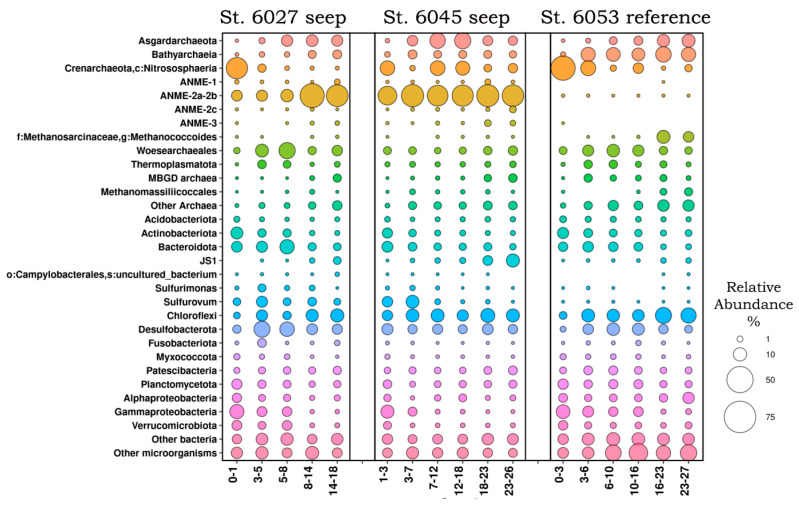
Microbial community composition in the surface horizons of the bottom sediments collected at three Laptev Sea stations (6027, 6045, 6053) according to the results of high-throughput sequencing of the 16S rRNA gene fragments.

**Figure 6 microorganisms-11-00250-f006:**
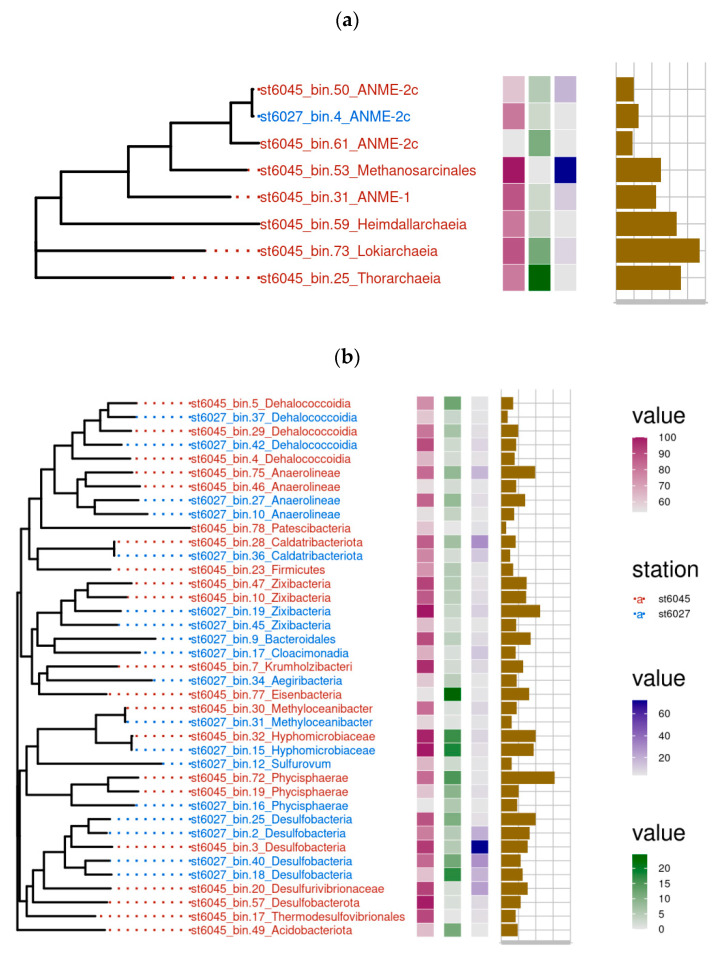
The main characteristics of obtained MAGs of archaea (**a**) and bacteria (**b**) at stations 6027 and 6045. First column—completeness of the assembled MAG (%); second column—contamination (%); third column—average sequencing coverage (fold); fourth column—MAG size (one cell—1 Mbp).

**Figure 7 microorganisms-11-00250-f007:**
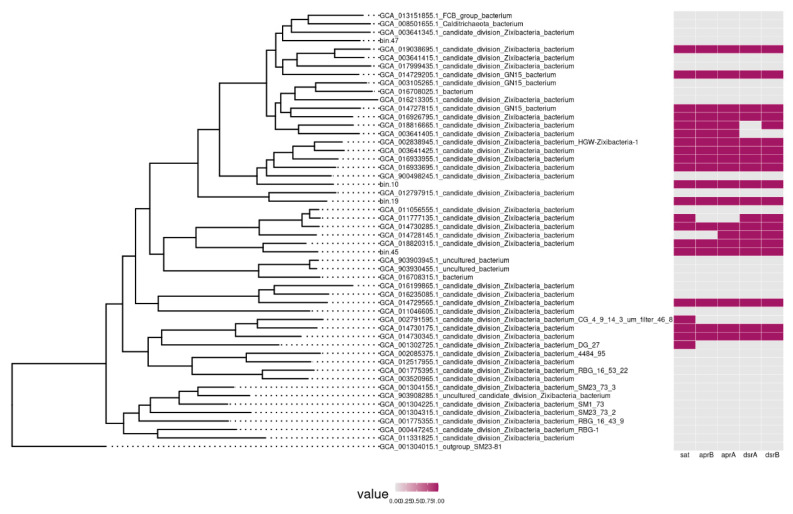
Phylogenetic tree of the phylum *Zixibacteria*. The presence of genes for the dissimilation pathway of sulfate reduction is shown to the right of the tree.

**Figure 8 microorganisms-11-00250-f008:**
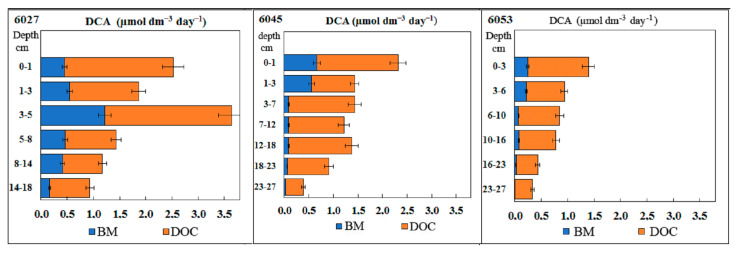
Rates of dark CO_2_ assimilation (DCA, µmol dm^−3^ day^−1^) in the Laptev Sea shelf sediments. CO_2_ carbon assimilation into organic matter of microbial biomass (BM) and dissolved organic carbon (DOC).

**Figure 9 microorganisms-11-00250-f009:**
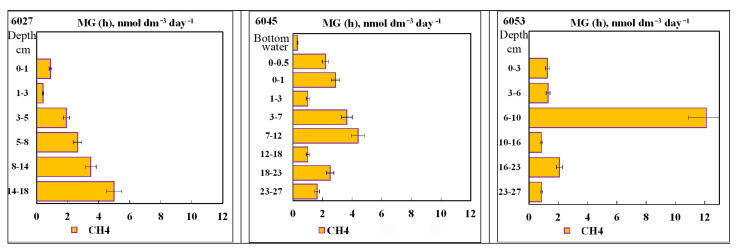
Rates of autotrophic (hydrogenotrophic) methanogenesis (MGh), nmol dm^−3^ day^−1^) in the Laptev Sea shelf sediments.

**Figure 10 microorganisms-11-00250-f010:**
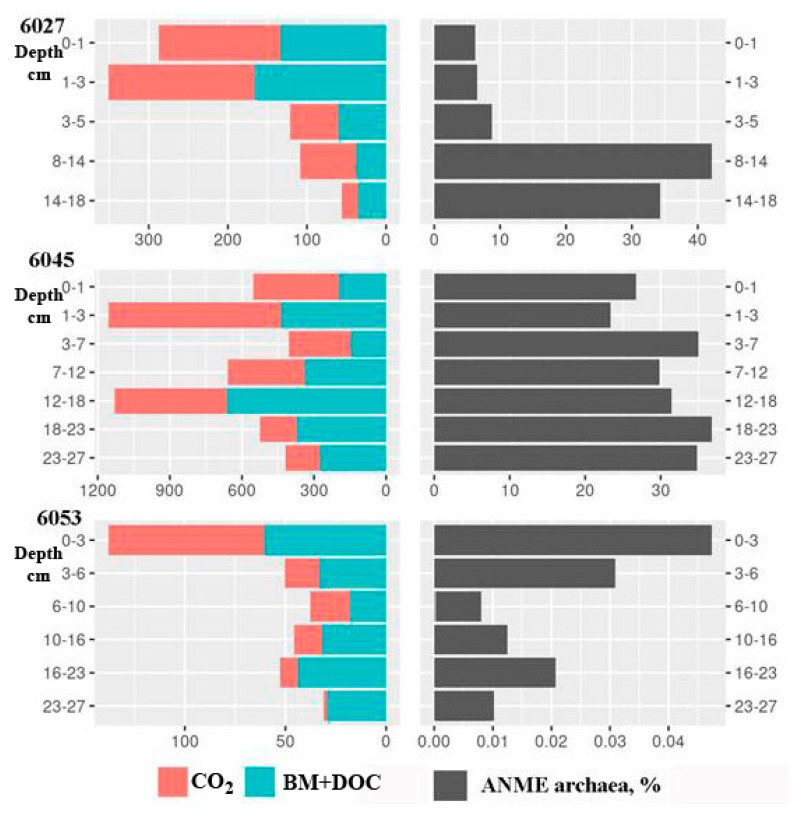
Rates of methane oxidation (MO, nmol dm^−3^ day^−1^) in the Laptev Sea shelf sediments. Methane carbon oxidation to CO_2_ and assimilation into organic matter of microbial biomass and dissolved organic carbon (BM + DOC).

**Figure 11 microorganisms-11-00250-f011:**
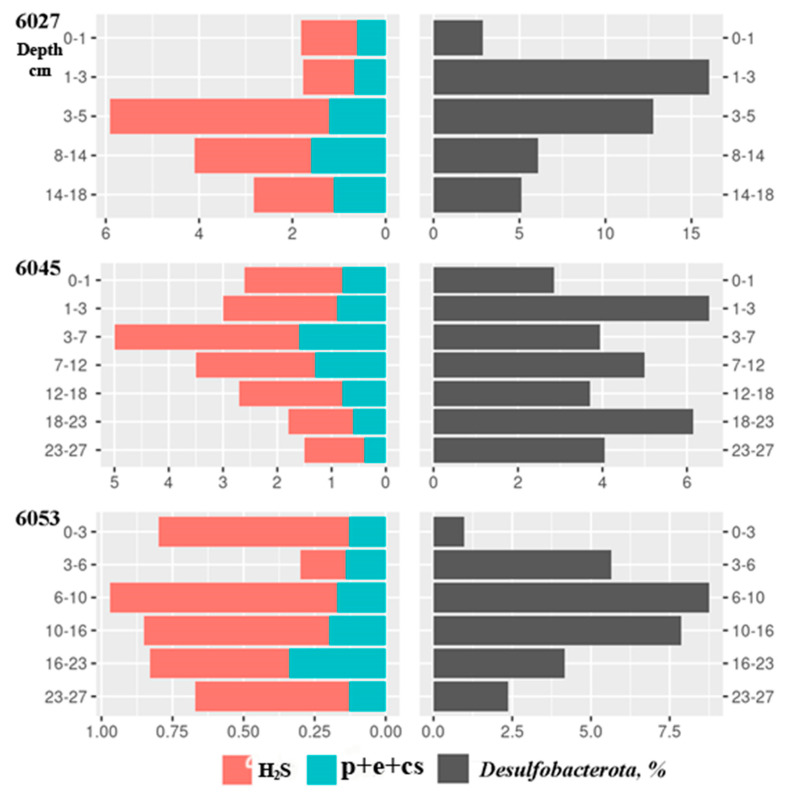
Sulfate reduction rates (SR, µmol dm^−3^ day^−1^) in the Laptev Sea shelf sediments. Reduction of sulfate sulfur to sulfide (H_2_S), reduction of pyrite sulfur to S^0^, and reduction of elemental sulfur and cellular sulfur (pyrite+elemetal+crystalline sulfur).

**Table 2 microorganisms-11-00250-t002:** Alpha diversity indices in the sediment samples from stations 6027, 6045, and 6053.

Sample	chao1	shannon_e
st. 6027 0–1 cm	960.5	5.04
st. 6027 3–5 cm	915.7	5.13
st. 6027 5–8 cm	791.1	5.12
st. 6027 8–14 cm	553.9	3.66
st. 6027 14–18 cm	695.8	4.38
st. 6045 0–1 cm	810.2	4.40
st. 6045 1–3 cm	871.6	4.73
st. 6045 3–7 cm	582.0	4.11
st. 6045 7–12 cm	557.6	3.90
st. 6045 12–18 cm	514.6	3.88
st. 6045 18–23 cm	711.7	4.78
st. 6045 23–26 cm	660.3	4.58
st. 6053 0–3 cm	1057.6	4.67
st. 6053 3–6 cm	1224.9	5.57
st. 6053 6–10 cm	1248.3	5.76
st. 6053 10–16 cm	994.0	5.29
st. 6053 16–23 cm	848.4	4.93
st. 6053 23–27 cm	755.4	4.78

**Table 3 microorganisms-11-00250-t003:** Carbon isotopic composition of organic matter (δ^13^C_org_) and methane carbon (δ^13^C-CH_4_) in the sediments of three Laptev Sea stations.

Station	Horizon	δ^13^C_org_	δ^13^C-CH_4_
6027	0–1	−27.7	*
Sep	1–3	−27.0	*
	3–5	−27.3	*
	5–8	−28.7	*
	8–14	−30.7	*
	14–18	−28.0	*
	Average	−28.2	
6045	0–1	−29.8	*
Seep	1–3	−30.2	*
	3–7	−27.8	−77.8
	7–12	−30.5	−81.5
	12–18	−28.4	−98.9
	18–23	−28.7	−80.5
	23–27	−31.8	−77.6
	Average	−29.6	−83.3
6053	0–3	−26.4	*
Reference	3–6	−27.8	*
	6–10	−26.1	*
	10–16	−27.2	*
	16–23	−27.0	*
	23–27	−25.9	*
	Average	−26.7	

* Methane amount was insufficient for reliable assessment.

**Table 4 microorganisms-11-00250-t004:** Comparison of Laptev Sea methane seeps to previously described methane-containing marine sediments, cold methane seeps, and the sediments with background methane content.

Study Area	[CH_4_] µmol dm^−3^	MG nmol dm^−3^ day^−1^	MO (AOM) nmol dm^−3^ day^−1^	SR µmol dm^−3^ day^−1^	δ^13^C-C_org_ ‰	References
HMMV (Barens Sea)	More 3000	6 ÷ 45	2.0 ÷ 75.8 × 10^3^	5.9 ÷ 394		[3,46]
Vestnesa Ridge pockmarks	More 3000	2.2 ÷ 75	7.2 ÷ 38 × 10^3^	0.6 ÷ 512		[46]
Black Sea sediment P817	80 ÷ 150		400 ÷ 700 × 10^3^	1400 ÷ 2100		[47], cited from [35]
Hikurangi margin, *Beggiatoa* site 315 (New Zealand)	15 ÷ 360		100 ÷ 500 × 10^3^	100 ÷ 1200		[35]
Gulf of Mexico cold seeps	To 2000		To 500 × 10^3^	To 5800	−26 ÷ −23	[34]
Gulf of Mexico Low seepage control	To 8.0		To 1.5 × 10^3^	To 7.8	−25	[34]
Sediments of South China Sea	2 ÷ 10	2 ÷ 29.6	0.4 ÷ 1.4 × 10^3^	0.01 ÷ 0.6		[48]
Laptev Sea methane seep	220 ÷ 539	20 ÷ 55	0.8 ÷ 4.2 × 10^3^	0.4 ÷ 48	−32.4 ÷ −29.2	[49]
Laptev Sea Low seepage control	0.1 ÷ 2.6	2 ÷ 8	22 ÷ 75	0.02 ÷ 0.8	−28.5–−26÷ 5	[49]
Kara Sea sediments Yamal sector	3.5 ÷ 20.5	0.8 ÷ 9.0	9.2 ÷ 103	0.46 ÷ 2.21	−27.5 ÷ −22.5	[18]
Northern part of the Kara Sea	0.02 ÷ 0.3	2.2 ÷ 7.5	0.2 ÷ 15	0.4 ÷ 2.2	−25.7 ÷ −21.5	[50]
Chukshi Sea sediments	0.18 ÷ 1.45	0.04 ÷ 0.85	15 ÷ 140	0.08 ÷ 1.8	−24.2 ÷ −21.7	[51]
Northern part of the Barents Sea	0.2 ÷ 9.5		21 ÷ 230	0.3 ÷ 2.8		[52]
Laptev Sea methane seeps	14 ÷ 37	0.4 ÷ 5.0	0.55 ÷ 1.2 × 10^3^	1.5 ÷ 6.0	−31.8 ÷ −27.8	Present work
Laptev Sea Low seepage control	0.15 ÷ 1.8	0.4 ÷ 12.0	30 ÷ 130	0.3 ÷ 1.0	−27.8 ÷ −25.9	Present work

## Data Availability

The raw data generated from the 16S rRNA gene and metagenomes sequencing were deposited in the NCBI Sequence Read Archive (SRA) and are available via the BioProject PRJNA679168.

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
