# Peer review of "Biogeochemical Activity of Methane-Related Microbial Communities in Bottom Sediments of Cold Seeps of the Laptev Sea"

_microorganisms, 2023, doi:10.3390/microorganisms11020250_

Round 1

Reviewer 1 Report

The authors present a variety of data about the composition, diversity of microbial communities in the sediments on the Laptev Sea shelf in the seep area and the background area. There are geochemical characteristics, the genesis and concentrations of methane in the studied cores, the rates of oxidation and methane formation and sulfate reduction processes. , the processes dominating in these deposits, and the microbial communities associated with them are measured. The analysis of these data made it possible to discuss the metabolic pathways and the significance of individual taxa of microorganisms, inhabiting in the specific environmental conditions. Of particular note is the use of metagenomic studies for the analysis of microbial communities as confirmation of the ability of some microorganisms to provide various processes. The diversity of the received information is a big plus of the this paper. Data of this level for the Arctic regions are very few, and the proposed work expands our knowledge in one of the northern seas.

Main note.

Section 2. Methods for analyzing and assembling genomes from MAG are not presented, but only methods for analyzing amplicons of the 16 S rRNA gene. The data in the results (3.3.) does not allow us to evaluate the percentage of contamination, the complete of the assembly, the program for the assembly, etc.

There is some dissonance in the presentation of this data. Is the diversity in the two samples by with the analysis of 16S rRNA amplicons differ in comparison with the analysis of this gene in MAGs?

Are you detected sequences of Zixibacteria in amplicon libraries, or were they in minor amounts and not shown in Fig. 5. To what extent their genomes are assembled and whether they differ from similar genomes from other sea ecosistems. It seems to me that the connection between these two approaches is not fully disclosed in the discussion.

L 525-527. Unclear what do you used - 16S rRNA gene amplicon analysis or MAG analysis - for concluded significant similarity significant similarities between microbial communities of the underwater methane seeps located in geographically remote areas of the Global Ocean.

Minor remarks

L 314 3.2. Composition of the Microbial Community of the Surface Layer of the Sediments  

The communities were studied in several layers of three cores, and not only in surface.

L 66-67 – duplicate of the words

Figure caption 6. Which genomes were compared,  from the obtained MAGs? What does value mean?

Figures 8 and 10.- change the font size

Figure 11 - poor quality drawings

Author Response

The authors present a variety of data about the composition, diversity of microbial communities in the sediments on the Laptev Sea shelf in the seep area and the background area. There are geochemical characteristics, the genesis and concentrations of methane in the studied cores, the rates of oxidation and methane formation and sulfate reduction processes. , the processes dominating in these deposits, and the microbial communities associated with them are measured. The analysis of these data made it possible to discuss the metabolic pathways and the significance of individual taxa of microorganisms, inhabiting in the specific environmental conditions. Of particular note is the use of metagenomic studies for the analysis of microbial communities as confirmation of the ability of some microorganisms to provide various processes. The diversity of the received information is a big plus of the this paper. Data of this level for the Arctic regions are very few, and the proposed work expands our knowledge in one of the northern seas.

We thank the reviewers for the positive evaluation of our research.

Main note.

Section 2. Methods for analyzing and assembling genomes from MAG are not presented, but only methods for analyzing amplicons of the 16 S rRNA gene. The data in the results (3.3.) does not allow us to evaluate the percentage of contamination, the complete of the assembly, the program for the assembly, etc. Answer: We added description of the methods for analyzing and assembling genomes.

There is some dissonance in the presentation of this data. Is the diversity in the two samples by with the analysis of 16S rRNA amplicons differ in comparison with the analysis of this gene in MAGs? Answer: Since 16S rRNA genes are usually present in genomes in several copies, they are assembled into separate contigs and are not included in MAGs during binning. Therefore, the vast majority of MAGs do not contain 16C rRNA genes. To analyze the composition of communities, only data on the 16S rRNA amplicons can be used.

Are you detected sequences of Zixibacteria in amplicon libraries, or were they in minor amounts and not shown in Fig. 5. To what extent their genomes are assembled and whether they differ from similar genomes from other sea ecosistems. It seems to me that the connection between these two approaches is not fully disclosed in the discussion. Answer: In amplicon libraries we detected sequences of Zixibacteria in minor amounts (less than 0,5%). In Figure 5 they are shown as “other bacteria”. Zixibacteria Bin19, was assembled with 100% completeness.

L 525-527. Unclear what do you used - 16S rRNA gene amplicon analysis or MAG analysis - for concluded significant similarity significant similarities between microbial communities of the underwater methane seeps located in geographically remote areas of the Global Ocean.

Answer: We used 16S rRNA gene amplicon analysis.

Minor remarks

L 314 3.2. Composition of the Microbial Community of the Surface Layer of the Sediments  

The communities were studied in several layers of three cores, and not only in surface.

Answer: We changed to “Composition of the Microbial Community of the Sediments”

L 66-67 – duplicate of the words . Corrected

Figure caption 6. Which genomes were compared,  from the obtained MAGs? What does value mean? Answer: This figure presents the main characteristics of the obtained MAGs. We modified the legend.

Figures 8 and 10.- change the font size

  The font size in the figures is increased

Figure 11 - poor quality drawings Answer: The figure was changed

Reviewer 2 Report

This study investigated biogeochemical activity of microbial communities that related to methane generation/consumption in bottom sediments of cold seeps in the Laptev Sea, the results are very promising. However, more discussion should be added in the results section, and only one sample was collected in each site, which may increase the uncertainty or human-related error. I would recommend the author revise the manuscript before publication. The detailed comments are as below.

Specific comments:

1.       Overall comments: in some figures, e.g. K coefficient at site 6027 in the 2-4 cm horizon (Figure 4), rates of DCA at site 6027 in the 3-5 cm depth (figure 8), there is a huge jump among the data, which probably because data was from only one sample at each site and that could increase the uncertainty. Therefore, I would suggest collecting more samples at each site to reduce the error in your next study. If no more samples can be added in this study, more explanations should be added in the text for these data.

2.       Line 21, please add the full name of ANME.

3.       Line 38, please remove the bracket between “of” and “the” and rephrase the first sentence.

4.       Line 139, there is no information about 16S rRNA gene reads in Table 1, please update the table number. There are same issues in other parts, please check and update them all together.

5.       Please keep the font size the same throughout the whole manuscript.

6.       As I mentioned before, please add more discussions for those jump data, K coefficient at site 6027 in the 2-4 cm horizon (Figure 4), rates of DCA at site 6027 in the 3-5 cm depth (figure 8), rates of MGh at site 6053 in the 6-10 cm depth.

7.       Line 400, “should b” should be replaced by “should be”.

8.       In Table 4, please replace “÷” to “–” and add “×” between number and “10^3”.

9.       Replace “Present publication” with “this study” or “present study” would be better.

Author Response

This study investigated biogeochemical activity of microbial communities that related to methane generation/consumption in bottom sediments of cold seeps in the Laptev Sea, the results are very promising. However, more discussion should be added in the results section, and only one sample was collected in each site, which may increase the uncertainty or human-related error. I would recommend the author revise the manuscript before publication. The detailed comments are as below.

 We thank the reviewer for this comment. The results of analysis of three sediment cores were used in our study. The same set of radiotracer experiments was performed for each sediment layer of each core. All analyses were carried out in triplicate. The article presents experimental data for each sediment layer. Marine sediments are known to contain thin layers differing significantly from the upper and lower ones. The factors responsible for existence of such layers are not always clear, and only conjectural explanations may be proposed. This is certainly the case for the DCA in the 3-5 cm horizon of st. 6027 and for the MG(h) value in the 6-10 cm horizon of station 6053. In our opinion, publication of the real experimental data makes it possible to illustrate the scattering of the data on the rates of microbial processes in the sediment layers.

Specific comments:

1.       Overall comments: in some figures, e.g. K coefficient at site 6027 in the 2-4 cm horizon (Figure 4), rates of DCA at site 6027 in the 3-5 cm depth (figure 8), there is a huge jump among the data, which probably because data was from only one sample at each site and that could increase the uncertainty. Therefore, I would suggest collecting more samples at each site to reduce the error in your next study. If no more samples can be added in this study, more explanations should be added in the text for these data. answer: As recommended by the reviewer, data from one sample with an unusually high value of the K coefficient was excluded from Figure 4. According to the reviewer's recommendations, some comments were added to Results 3.6.

2.       Line 21, please add the full name of ANME.

 Corrected

3.       Line 38, please remove the bracket between “of” and “the” and rephrase the first sentence. Done

4.       Line 139, there is no information about 16S rRNA gene reads in Table 1, please update the table number. There are same issues in other parts, please check and update them all together. Information on sequences has been added to Table 1.

5.       Please keep the font size the same throughout the whole manuscript Corrected

6.       As I mentioned before, please add more discussions for those jump data, K coefficient at site 6027 in the 2-4 cm horizon (Figure 4), rates of DCA at site 6027 in the 3-5 cm depth (figure 8), rates of MGh at site 6053 in the 6-10 cm depth.

 According to the reviewer's recommendations, some comments were added to Results 3.6

7.       Line 400, “should b” should be replaced by “should be Corrected

8.       In Table 4, please replace “÷” to “–” and add “×” between number and “10^3”. Corrected

9.       Replace “Present publication” with “this study” or “present study” would be better. Corrected

Round 2

Reviewer 2 Report

Thanks for your revision, the quality of this manuscript improved a lot.